# The Current Role of the sFlt-1/PlGF Ratio and the Uterine–Umbilical–Cerebral Doppler Ultrasound in Predicting and Monitoring Hypertensive Disorders of Pregnancy: An Update with a Review of the Literature

**DOI:** 10.3390/children10091430

**Published:** 2023-08-22

**Authors:** Cristian Nicolae Chirilă, Claudiu Mărginean, Paula Maria Chirilă, Mirela Liana Gliga

**Affiliations:** 1Department of Internal Medicine-Nephrology, Doctoral School, George Emil Palade University of Medicine, Pharmacy, Science and Technology of Târgu Mureș, 540142 Târgu Mureș, Romania; cristian.chirila@umfst.ro (C.N.C.); mirela.gliga@umfst.ro (M.L.G.); 2Department of Nephrology, Mures Clinical County Hospital, 540103 Târgu Mureș, Romania; 3Department of Obstetrics and Gynecology 2, George Emil Palade University of Medicine, Pharmacy, Science and Technology of Târgu Mureș, 540142 Târgu Mureș, Romania; 4Department of Obstetrics and Gynecology, Mures Clinical County Hospital, 540057 Târgu Mureș, Romania; 5Department of Endocrinology, Mures Clinical County Hospital, 540142 Târgu Mureș, Romania; paulachirila@yahoo.com; 6Diaverum Dialysis Centre, 540487 Târgu Mureș, Romania

**Keywords:** prediction, pre-eclampsia, gestational hypertension, sFlt-1/PlGF ratio, uterine Doppler, fetal growth restriction

## Abstract

Regarding the hypertensive disorders of pregnancy, pre-eclampsia (PE) remains one of the leading causes of severe and life-threatening maternal and fetal complications. Screening of early-onset PE (<34 weeks of pregnancy), as well as late-onset PE (≥34 weeks), shows poor performance if based solely on clinical features. In recent years, biochemical markers from maternal blood—the pro-angiogenic protein placental growth factor (PlGF) and the antiangiogenic protein soluble FMS-like tyrosine kinase 1 (sFlt-1)—and Doppler velocimetry indices—primarily the mean uterine pulsatility index (PI), but also the uterine resistivity index (RI), the uterine systolic/diastolic ratio (S/D), uterine and umbilical peak systolic velocity (PSV), end-diastolic velocity (EDV), and uterine notching—have all shown improved screening performance. In this review, we summarize the current status of knowledge regarding the role of biochemical markers and Doppler velocimetry indices in early prediction of the onset and severity of PE and other placenta-related disorders, as well as their role in monitoring established PE and facilitating improved obstetrical surveillance of patients categorized as high-risk in order to prevent adverse outcomes. A sFlt-1/PlGF ratio ≤ 33 ruled out early-onset PE with 95% sensitivity and 94% specificity, whereas a sFlt-1/PlGF ≥88 predicted early-onset PE with 88.0% sensitivity and 99.5% specificity. Concerning the condition’s late-onset form, sFlt-1/PlGF ≤ 33 displayed 89.6% sensitivity and 73.1% specificity in ruling out the condition, whereas sFlt-1/PlGF ≥ 110 predicted the condition with 58.2% sensitivity and 95.5% specificity. The cut-off values of the sFlt-1/PlGF ratio for the screening of PE were established in the PROGNOSIS study: a sFlt-1/PlGF ratio equal to or lower than 38 ruled out the onset of PE within one week, regardless of the pregnancy’s gestational age. The negative predictive value in this study was 99.3%. In addition, sFlt-1/PlGF > 38 showed 66.2% sensitivity and 83.1% specificity in predicting the occurrence of PE within 4 weeks. Furthermore, 2018 ISUOG Practice Guidelines stated that a second-trimester mean uterine artery PI ≥ 1.44 increases the risk of later PE development. The implementation of a standard screening procedure based on the sFlt-1/PlGF ratio and uterine Doppler velocimetry may improve early detection of pre-eclampsia and other placenta-related disorders.

## 1. Introduction

The way in which pregnancy may cause hypertension or aggravate pre-existing hypertension has been proven challenging to understand, although decades of sustained obstetrical research has focused on explaining the pathophysiology [1,2].

It is well known that hypertensive disorders affect 7–10% of all pregnancies worldwide, representing one of the three main conditions responsible for severe maternal and fetal morbidity and mortality, alongside hemorrhage and infection [1].

Many classification systems have been proposed for the hypertensive disorders of pregnancy, with the first one being introduced in 1972. The actual classification was elaborated by the American College of Obstetricians and Gynecologists in 2013. It provides evidence-based recommendations for clinical practice, though it keeps the basic principles of the previous classifications. It describes four types of hypertension in pregnancy, such as gestational hypertension, pre-eclampsia and eclampsia syndrome, chronic hypertension (of any cause), and chronic hypertension with superimposed pre-eclampsia [2].

Diagnostic criteria for gestational hypertension include at least two maternal blood pressure values of over 140/90 mmHg measured at rest after 20 weeks of gestation in a previously normotensive patient, with no evidence of accompanying proteinuria.

According to World Health Organization (WHO) data, the incidence of pre-eclampsia is currently rising, affecting between 2 and 10% of pregnancies worldwide [3]. It is regarded as a syndrome due to its multiorgan involvement. Specific criteria must be met to establish the diagnosis—the classic diagnosis criteria include hypertension appearing after 20 weeks of gestation and proteinuria. The occurrence of proteinuria is a consequence of systemic endothelial dysfunction that causes, among other signs, abnormal protein excretion. Proteinuria may be defined as 24-hour urine excretion of protein content of over 300 mg or a protein/creatinine ratio ≥ 0.3 and persistent protein (dipstick 1+ or >30 mg/dL) in the morning urine sample [1,2].

However, as it is a multisystemic disease, it is noticeable that there are some cases in which pre-eclampsia may occur in the absence of proteinuria. In these cases, the diagnosis includes elevated blood pressure ≥ 140/90 mmHg, alongside multiorgan signs, such as thrombocytopenia (platelet count < 100,000/microliter), impaired liver function (transaminase levels in the blood elevated to twice the normal values), new onset of renal insufficiency (serum creatinine > 1.1 mg/dL or doubling of creatinine levels without previous renal impairment), and new onset of visual and cerebral impairments, i.e., headaches, scotomas, pulmonary edema [1,2].

Two forms of pre-eclampsia have been described: early-onset (<34 weeks) and late-onset (≥34 weeks) pre-eclampsia [4]. Moreover, pre-eclampsia may be categorized as severe in women who meet the basic diagnosis criteria if certain indicators of disease severity are present, such as blood pressure of >160/110 mmHg measured at rest, the above-mentioned multiorgan signs, or epigastric or right upper quadrant pain. Thrombocytopenia, as a trigger of platelet aggregation and microangiopathic hemolysis, is generally regarded as an indicator of worsening pre-eclampsia, while the occurrence of headaches and visual impairment are predictive symptoms of eclampsia [2].

## 2. Pre-Eclampsia Screening

Pre-eclampsia may increase the risk of maternal, as well as fetal, morbidity and mortality [2]. Adverse pregnancy outcomes consist of uncontrollable severe hypertension, eclampsia (a life-threatening hypertensive disorder), stroke, myocardial infarction, pulmonary edema, acute respiratory distress syndrome, HELLP syndrome, disseminated intravascular coagulation, severe renal failure, retinal lesions, abruptio placentae, intrapartum fetal demise, fetal growth restriction, and pre-term birth, as well as long-term neonatal morbidities, such as cerebral palsy and chronic pulmonary hypertension of the newborn [5,6,7]. The risk of patients experiencing these complications is greater in severe and early- onset forms of pre-eclampsia [8].

Certain anamnestic factors, as well as clinical factors, are known to increase the risk of pre-eclampsia. Anamnestic factors include having a first-degree relative with medical history of the disorder, previous pregnancy complicated by pre-eclampsia, and personal history of thrombophilia, while predisposing clinical factors are represented by multifetal gestation—triplet gestation carries an increased risk compared to twin gestation—as well as in vitro fertilization, advanced maternal age (over 40 years), diabetes mellitus type I or II, obesity with a BMI > 35 kg/m^2^, pre-existing chronic hypertension or chronic kidney disease, autoimmune disease such as systemic lupus erythematosus, significantly increased serum uric acid levels (more than 1.5 times higher than normal after 20 weeks of gestation), and the more recently described serum uric acid-to-creatinine ratio, which is significantly higher in all three trimesters among pregnant women who later develop pre-eclampsia [2,9,10,11].

However, it is of the utmost importance to note that the vast majority of pre-eclampsia cases affect healthy nulliparous women with none of the clinical factors mentioned above. Therefore, the prediction of pre-eclampsia using clinical factors has shown poor predictive value, with a 37% detection rate for early-onset pre-eclampsia and a 29% detection rate for late-onset pre-eclampsia [2].

As pre-eclampsia remains one of the main causes of severe and life-threatening maternal and fetal complications, there is a need for more reliable and accurate prediction tests. In order to early predict or diagnose the condition, biochemical markers, as well as Doppler velocimetry indices, have been evaluated in multiple studies in recent years. More than 10,000 articles concerning screening strategies for pre-eclampsia have been published on PubMed, illustrating the great interest in this topic [12].

## 3. Prediction and Clinical Management of Pre-Eclampsia and Other Adverse Pregnancy Outcomes Using the sFlt-1/PlGF Ratio

The pathogenesis of pre-eclampsia is complex and not entirely explained. Alteration of the normal structure of maternal spiral arteries, the lumen of which becomes abnormally narrow, leads to placental hypoperfusion and endothelial dysfunction. The subsequent placental dysfunction creates an imbalance between the abnormally increased levels of antiangiogenic factors and abnormally low levels of pro-angiogenic factors. Specifically, the roles played by two angiogenic factors—soluble fms-like tyrosine kinase 1 (sFlt-1), which is an antiangiogenic protein, and placental growth factor (PlGF), which is a pro-angiogenic protein—has been extensively researched in the recent literature, with relevant data generated regarding their ability to predict pre-eclampsia, as well as intrauterine growth restriction, pre-term delivery, and stillbirth [13]. While sFlt-1 acts as an antagonist of vascular endothelial growth factor, thus impairing normal vascular growth and proliferation, PlGF displays pro-angiogenic effects, enhancing the activity of vascular endothelial growth factor [14,15].

Increased sFlt-1 values and decreased PlGF levels in maternal blood can be used to predict later development of pre-eclampsia during pregnancy. The imbalance of the two angiogenic biomarkers typically predicts de novo pre-eclampsia and is not specific to chronic hypertension with superimposed pre-eclampsia [16]. sFlt-1 values start to increase 4–5 weeks before the onset of pre-eclampsia, whereas PlGF values begin to decrease 9–11 weeks before the clinical onset. Moreover, the sFlt-1/PlGF ratio had better accuracy than the individual markers used for the prognosis of the disease. The main reason for this variation is that PlGF alone may show such low values in severe or early-onset pre-eclampsia that they cannot be determined via commercial PlGF kits [17]. This ratio is regarded as an effective tool for physicians to use to enable early identification pregnancies at high risk of pre-eclampsia, thus enabling closer monitoring of these patients from an early stage of the pregnancy. The sFlt-1/PlGF ratio has been used to implement strategies for first-trimester screening of patients at risk of developing early-onset pre-eclampsia, who consequently receive prophylaxis with low-dose acetylsalicylic acid. Additionally, sFlt-1/PlGF ratio values are determined in the second trimester with the purpose of predicting late-onset pre-eclampsia, showing promising results [18].

## 4. Prediction and Clinical Management of Pre-Eclampsia and Other Adverse Pregnancy Outcomes Using Doppler Velocimetry Indices

As mentioned earlier in this paper, the pathogenesis of pre-eclampsia is complex and not entirely explained. Defective placentation leads to abnormal narrowing of the spiral arteriolar lumen, leading to impairment of the placental blood flow. Moreover, abnormal trophoblastic invasion of uterine vessels increases the resistance to flow within the uterine arteries, which may result in an abnormal waveform aspect, increased resistivity (RI), or pulsatility (PI) indices and the persistence of a unilateral or bilateral diastolic notch (Figure 1 right) [2,5].

Thus, a lot of emphasis has been put on the effectiveness of using uterine artery Doppler to screen pre-eclampsia. Uterine Doppler is regarded as a better predictor of early- onset pre-eclampsia than the late-onset form [2].

## 5. Uterine Artery Doppler—Technique and Reference Parameters

According to 2018 ISUOG Practice Guidelines, the pulsatility index should be the first parameter assessed in the context of pre-eclampsia screening. In the first trimester, mainly during the 11–14 weeks of gestation, uterine artery PI values > 90th centile predict 48% of early-onset pre-eclampsia cases and 26% of total pre-eclampsia cases. Besides PI, uterine artery notching, which is a marker of endothelial dysfunction, has also been assessed, but it appears to be relatively common in pregnancy (43% of normal pregnancies in the first trimester); therefore, mean uterine artery PI remains the gold-standard Doppler parameter in pre-eclampsia screening [12].

As in the first trimester, mean uterine artery PI should be assessed in the second trimester, with the purpose of predicting pre-eclampsia in high-risk women based on clinical factors. At 23 weeks, the 95th centile of the uterine artery PI obtained via transabdominal ultrasound has been established as 1.44. It decreases by 15% between 20 and 24 weeks and by <10% between 22 and 24 weeks. In transvaginal examination, uterine PI values are always higher than those measured transabdominally. The 95th centile of the uterine PI in the transvaginal approach has been reported to be 1.55 [12]. The normal Doppler values of the uterine artery PI in the first, second, and third trimesters are reported to be 1.84 ± 0.55, 1.07 ± 0.38, and 0.78 ± 0.23, respectively [19].

First-trimester transabdominal Doppler evaluation of the uterine artery is performed by obtaining a sagittal section of the uterus and cervix. Color flow should be activated, and the transducer is moved sideways until the uterine arteries are visualized on the left and right sides of the uterus and cervix. The sampling gate should be set at 2 mm and placed as close to the internal cervical as possible. The insonation angle should be maintained < 30°. A minimum of three identical consecutive waveforms should be recorded before PI measurement [12]. Second-trimester uterine artery Doppler evaluation using a transabdominal approach is performed by placing the 3.5-megahertz transducer in the right and left iliac fossae oriented toward the lateral walls of the uterus and the pelvis. The uterus and cervical canal should be visualized in a sagittal section. The location of the placenta should be noticed. After activation of the color flow mode, the transducer is moved sideways (without changing the medial angulation) until the uterine artery is observed as it crosses the external iliac artery. The sampling gate should be set at 2 mm and placed on the uterine artery at a distance of 1 cm distal from the point at which the external iliac artery crosses the uterine artery. The uterine artery is identified at the crossover with the right external iliac artery. The insonation angle should be maintained <50°. PI values are measured after recording at least three identical consecutive waveforms (Figure 1) [12,20].

## 6. Other Vascular Structures Evaluated in the Pregnancy Follow-Up—Doppler Technique and Reference Parameters

Color Doppler ultrasound (CDUS) is a non-invasive, radiation-free, and practical method of pregnancy surveillance. Umbilical and middle cerebral arteries (MCA) should be examined in both normal and pathological pregnancies, especially in those women affected by hypertensive disorders, in order to document any possible fetal sufferance as early as possible.

The second-trimester ultrasound evaluation of the umbilical artery consists of the following steps: after the visualization of an uncompressed loop of the umbilical cord, the color flow mode is activated, and the sampling gate should be set at 1–2 mm, to ensure that only the targeted section of the umbilical artery is examined; the angle of insonation should be kept as low as possible; the assessment should be performed in the absence of fetal movement or breathing; and after obtaining at least three consecutive identical waveforms, PI and RI values should be recorded [21]. In normal circumstances, the umbilical artery PI decreases from 1.27 to 0.69, whereas umbilical RI decreases from 0.75 to 0.6 due to the normal increase in diastolic blood flow during fetal development [22].

The fetal MCA is the cerebral vessel most easily assessed via ultrasound. The main part of the fetal cerebral blood flows through it. The two most commonly assessed parameters of MCA blood flow are the peak systolic velocity (PSV) and RI. However, the evaluation should also include the PI and cerebroplacental ratio (CPR). Doppler evaluation of the fetal MCA should start with a transverse section of the head at the level of the sphenoid bones. The thalamus and cavum septum pellucidum are also landmarks in an appropriate section. The color Doppler mode should be activated, and the MCA originating from circle of Willis should be visualized. MCA should be recorded along its entire length. The sampling gate should be set at 1–2 mm and placed as close to the origin of MCA from the circle of Willis as possible. The angle of insonation should be maintained < 30°. The evaluation has to be performed in the absence of fetal movement or breathing. After obtaining at least three minimum waveforms, PSV is calculated based on the highest waveform, while PI and RI are usually automatically assessed via the device. CPR is obtained by dividing the Doppler parameters RI and PI of the MCA by those of the umbilical arteries. CPR values < 1 are considered to be abnormal. PSV values normally increase with gestational age. To evaluate a pathological increase in PSV, conversion to values of multiples of median (MoM) may be required [21].

The aim of this paper is to perform a comprehensive review of the current research regarding the roles of the biochemical markers sFlt-1 and PlGF and Doppler velocimetry in the prediction and monitoring of early- and late-onset preeclampsia, gestational hypertension, and other placenta-related disorders. At the same time, this paper also highlights the added value brought by the sFlt-1/PlGF ratio or uterine Doppler velocimetry markers in the prediction and management of hypertensive disorders of pregnancy among high-risk women compared to judgement only based on clinical criteria. For this purpose, an extensive analysis of articles published between 2009 and 2023 in the PubMed database has been performed, including the following search terms: “prediction”, “preeclampsia”, “fetal growth restriction”, “sFlt-1/PlGF”, and “uterine Doppler”. The present paper primarily focuses on publications from the last six years.

## 7. Relevant Data from the Recent Literature Regarding the Value of the sFlt-1/PlGF Ratio

The assessment of the sFtl-1/PlGF ratio performance in the prediction of hypertensive disorders in pregnancy proved to be a central focus of recent studies. Among these foci, some of the studies only evaluated the reliability of the two biomarkers for the screening and monitoring of preeclampsia, while others took into account sFlt-1/PlGF’s role in predicting intrauterine growth restriction as a complication of pregnant patients with preeclampsia. Meanwhile, two studies focused on sFlt-1/PlGF ratio’s role as a reliable parameter that enables the accurate delineation of preeclampsia from the other hypertensive disorders.

Verlohren [23] conducted one of the first studies to implement gestational age-dependent cut-offs for the sFlt-1/PlGF ratio as a prognostic tool for PE: sFlt-1/PlGF ≤ 33 ruled out early-onset pre-eclampsia with 95% sensitivity and 94% specificity, whereas sFlt-1/PlGF ≥ 88 predicted early-onset pre-eclampsia with 88.0% sensitivity and 99.5% specificity. Concerning the late-onset form, sFlt-1/PlGF ≤ 33 displayed 89.6% sensitivity and 73.1% specificity in ruling out the condition, whereas sFlt-1/PlGF ≥ 110 predicted the condition with 58.2% sensitivity and 95.5% specificity.

The PROGNOSIS study (Prediction of Short-Term Outcome in Pregnant Women with Suspected Pre-Eclampsia Study), which was a major prospective clinical trial, established cut-off values of the sFlt-1/PlGF ratio for the screening of preeclampsia: a sFlt-1/PlGF ratio ≤ 38 ruled out the onset of pre-eclampsia within one week, regardless of the gestational age. The negative predictive value in this study was 99.3%. In addition, sFlt-1/PlGF > 38 showed 66.2% sensitivity and 83.1% specificity in predicting the occurrence of pre-eclampsia within 4 weeks [24]. The cut-off values established in the PROGNOSIS study proved a similar performance of prediction in the PROGNOSIS Asia study, where the negative predictive value of sFlt-1/PlGF ≤ 38 in terms of ruling out pre-eclampsia within one week was 98.6%. Furthermore, significantly higher sFlt-1/PlGF values could be noticed in case of patients who developed pre-eclampsia at any time during pregnancy compared to the control (45.5 vs. 6.6), among patients who developed fetal adverse outcomes compared to the control within 1 and 4 weeks (148.9 vs. 7.4 and 86.9 vs. 6.3, respectively), and in women with pre-term delivery (<37 weeks) compared to term delivery among both early-onset and late-onset preeclampsia. sFlt-1/PlGF above the cut-off of 38 was also a predictive marker of shorter pregnancy duration, though it was independent of preeclampsia diagnosis [25].

Moreover, Stepan et al. [26] established additional cut-off values requiring intensive obstetrical monitoring: a sFlt-1/PlGF ratio > 85 is suggestive of a high risk of early-onset pre-eclampsia or placenta-related disorders, while a sFlt-1/PlGF ratio > 110 may be used as a predictor of late-onset pre-eclampsia or placenta-related disorders. sFt-1/PlGF ratio values between 38 and 85 rule out current pre-eclampsia but indicate the pregnancy carries high risk of early-onset preeclampsia development within the next four weeks, whereas values between 38 and 110 increase the risk of developing late-onset preeclampsia within the next four weeks.

Zeisler et al. [27] used a post hoc analysis of the PROGNOSIS study to assess whether the sFlt-1/PlGF ratio cut-off ≤ 38, which has been previously known to rule out pre-eclampsia onset within one week, could effectively rule out pre-eclampsia in two, three, and four weeks after measurement. For this purpose, the sFlt-1/PlGF ratio was determined four times during pregnancy: at the baseline and after two, three, and four weeks. The median difference in sFlt-1/PlGF values was significantly higher (*p* < 0.001) at 2, 3 and 4 weeks in the case of patients who developed pre-eclampsia compared to the control group. For example, the median difference at two weeks compared to baseline was 1.40 for the control group, whereas women who developed pre-eclampsia recorded a median difference of 21.22. The authors obtained excellent negative predictive values (97.9% at 2 weeks, 95.7% at 3 weeks, and 94.3% at 4 weeks), thus proving that the sFlt-1/PlGF ratio threshold ≤ 38 is a reliable indicator of ruling out pre-eclampsia for up to four weeks in the case of women with suspected pre-eclampsia based on protocol-defined criteria. Moreover, they proved that repeated determination of the biomarkers after 2, 3, and 4 weeks allowed better risk stratification and helped clinicians in the decision-making process.

Cerdeira, during the INSPIRE study [28], used a sFlt-1/PlGF cut-off value of 38 to classify a low risk (≤38) and high risk (>38) of developing pre-eclampsia within the next week after determination of the markers. Authors showed that sFlt-1/PlGF significantly improved the precision in predicting pre-eclampsia compared to clinical practice alone (100% sensitivity vs. 83% sensitivity and 100% negative predictive value vs. 97.8% negative predictive value, respectively). The post hoc analysis of the INSPIRE trial [29] proved that the cut-off of 38 was modest in predicting pre-eclampsia occurrence within 4 weeks, but a sFlt-1/PlGF ratio ≥ 85 had a significant effect on the precision of ruling-in the condition within the next 4 weeks, having a 71.4% positive predictive value.

Another study conducted by Caillon et al. [30], a prospective study including 67 women with ongoing pregnancy between 20 and 37 gestational weeks who presented at least one risk factor for pre-eclampsia found a significant difference (*p* = 0.01) between the mean sFlt-1/PlGF values of 32 ± 25 for women who did not develop the disease, whereas for women who later developed pre-eclampsia, the mean sFlt-1/PlGF values were 69 ± 13. Consequently, the authors proved that sFlt-1/PlGF can be applied to rule out pre-eclampsia in a specific population of high-risk patients, accurately separating high-risk patients who require intensive monitoring and high-risk patients for whom hospitalization was not necessary, despite having a risk factor. It is worth mentioning that three patients included in this study, whose sFlt-1/PlGF ratios were above 38, were not categorized as pre-eclamptic, despite presenting an atypical form of pre-eclampsia without proteinuria.

STEPS study, which was a major prospective study involving 729 patients at 10 sites in Spain and conducted by Perales et al. [31], also included women with one risk factor of pre-eclampsia, as did Caillon et al.. They found higher median sFlt-1/PlGF values at 20 weeks of pregnancy in women who developed early-onset pre-eclampsia (14.5) compared to the control group (7). Furthermore, the ratio values increased to 18.4 at 24 weeks and to 51.9 at 28 weeks in case of the patients who developed early-onset pre-eclampsia, whereas women from the control group displayed minimal change in median sFlt-1/PlGF values between 20 and 28 weeks, being <7. Resultantly, there was a significant difference (*p* < 0.001) in values at all three timepoints between women with early-onset pre-eclampsia and the control. Moreover, patients with early-onset pre-eclampsia showed significantly increased sFlt-1/PlGF at 20, 24, and 28 weeks compared to patients with late-onset pre-eclampsia or chronic and gestational hypertension. Only at 28 weeks of pregnancy could a significant statistical difference in the sFlt-1/PlGF ratio between women with late-onset pre-eclampsia and the control group be obtained. Perales et al. implemented a prediction model for early-onset pre-eclampsia centered on sFlt-1/PlGF values.

In a multicentric study, Diguisto [32] found significantly lower PlGF levels at 11–13 weeks among patients who later developed pre-eclampsia compared to the control (37.11 vs. 57.69 pg/mL, *p* < 0.001). On the other hand, no statistically significant differences in sFlt-1 values could be emphasized (*p* = 0.27). However, due to the differences in PlGF levels, the sFlt-1/PlGF ratio was also significantly increased among pregnancies complicated by pre-eclampsia (33.4 vs. 20.6, *p* < 0.001).

Additionally, sFlt-1/PlGF may be used to predict the severity of pre-eclampsia and certain adverse perinatal outcomes. Soundararajan et al. [33] conducted a study involving 50 high-risk third-trimester patients and categorized the prognoses into three classes based on sFlt-1/PlGF ratio values: low risk for values < 33, intermediate risk for values between 33 and 85, and high risk for values > 85. Patients with a sFlt-1/PlGF ratio > 85 carry a significant risk (*p* < 0.001) of developing severe pre-eclampsia (90.9% compared to 8% in case of women with sFlt-1/PlGF < 33) associated with pre-term birth (32.6 weeks compared to 37.4 weeks in case of women classified as being at low risk). Moreover, Tan et al. [34] showed that the combination of the sFlt-1/PlGF ratio and maternal risk factors displayed similar screening performance at 31–34 weeks for the prediction of delivery within the next four weeks due to pre-eclampsia as using sFlt-1/PlGF alone. On the other hand, combining sFlt-1/PlGF ratio with maternal risk factors showed superior performance in the prediction of delivery due to pre-eclampsia four weeks after the assessment.

Herraiz et al. [35] conducted an extensive study of more than 5000 singleton pregnancies, determining the sFlt-1/PlGF ratio at 24–28 weeks of pregnancy in case of women previously categorized as being at high-risk of developing pre-eclampsia based on clinical findings and uterine artery Doppler. They reported 100% sensibility and 80.6% specificity of the sFlt-1/PlGF ratio > 95th centile, when measured at 24–28 weeks, for the prediction of early-onset pre-eclampsia with intrauterine growth restriction. The authors found the sFlt-1/PlGF ratio ≥ 85 to be optimal at predicting pre-eclampsia or intrauterine growth restriction requiring delivery before 32 weeks. Beyond 32 weeks, sFlt-1/PlGF’s (measured at 24–48 weeks) ability to predict pre-eclampsia or intrauterine growth restriction decreased. However, the likelihood ratio remained five times higher if sFlt-1/PlGF values were > 95th centile.

In another study, Ciciu et al. [36] found the sFlt-1/PlGF ratio to be reliable in terms of determining the diagnosis of pre-eclampsia and distinguishing established between pre-eclampsia and uncomplicated gestational hypertension: mean sFlt-1/PlGF values in cases already carrying a diagnosis of pre-eclampsia were 209.2 (with a standard deviation of 138.77) compared to recorded mean values of 46.08 (standard deviation 17.37) in the gestational hypertension group and just 3.9 in the control group (standard deviation 0.3). The differences in the sFlt-1/PlGF ratio between the three groups were statistically significant (*p* < 0.001). In addition, statistically significant differences were recorded in the median sFlt-1/PlGF values between women with early-onset pre-eclampsia and women with early-onset gestational hypertension. Regarding the severity of pre-eclampsia, sFlt-1/PlGF ratio proved to be a reliable indicator of differentiating between the mild form (median ratio 77) and the severe form (median sFlt-1/PLGF ratio 303). Another interesting finding of the study is that based on sFlt-1/PlGF values, the risk of developing severe pre-eclampsia is 5625 times higher in patients with early-onset pre-eclampsia than those with late-onset pre-eclampsia.

Yang et al. [37] also stated that a distinction between pre-eclampsia and other forms of hypertensive disorders in pregnancy can be performed based on sFlt-1/PlGF values. In addition, authors found a ratio cut-off of 85 for severe pre-eclampsia with adverse outcomes, such as pre-term delivery. On the other hand, when Nikuei et al. [38] assessed the ability of sFlt-1/PlGF to differentiate between different forms of pre-eclampsia, they found no statistical difference in sFlt-1/PlGF levels between mild and severe forms of pre-eclampsia (*p* = 0.389) or the early- and late-onset forms of pre-eclampsia (*p* = 0.503). Significant differences regarding sFlt-1/PlGF values were obtained, however, between the early-onset form and the controls, as well as between the late-onset form and the controls (*p* < 0.001).

Leanos-Miranda [39] classified women who were already diagnosed with pre-eclampsia based on clinical factors into three groups based on the severity of angiogenic imbalance: no angiogenic imbalance if sFlt/PlGF ≤ 38, mild angiogenic imbalance if the ratio recorded values between 38 and 85, and severe angiogenic imbalance for sFlt-1/PlGF values ≥ 85. Rates of pre-term delivery, delivery within 14 days, and delivery of a baby small for its gestational age (SGA) infant were reported to be significantly higher among patients with severe angiogenic imbalance than patients with mild and no imbalance (*p* < 0.001), as well as higher among patients with mild imbalance than those with no imbalance (*p* ≤ 0.01). Furthermore, HELLP syndrome occurred only in women with severe imbalance. In another study, Baltajian [40] included patients diagnosed with pre-eclampsia and determined sFlt-1/PlGF values daily for the first three days, and then on a weekly basis until delivery. Thus, the authors established a relationship between sFlt-1/PlGF values and the number of days until delivery: for sFlt-1/PlGF values ≥ 85, the mean number of days until delivery was 6, while for sFl-1/PlGF values < 85, the mean number of days until delivery was 14 (*p* < 0.01). In addition, patients who presented adverse outcomes recorded significantly elevated median sFlt-1/PlGF values compared to those without adverse outcomes (205.9 vs. 47.5, *p* < 0.001).

Peguero [41] studied the relationship between longitudinal changes in the angiogenic proteins levels at admission and before delivery and the occurrence of adverse outcomes among patients with severe early-onset pre-eclampsia. The authors found that median longitudinal changes in sFlt-1 for pregnancies with confirmed early-onset pre-eclampsia were significantly increased among women who developed adverse outcome (1047 vs. 342 pg/mL/day or 8.2% vs. 2.6% daily increase, *p* = 0.04). Garcia-Manau [42] focused on the sFlt-1/PlGF ratio in the assessment of fetal growth restriction, discovering that median sFlt-1/PlGF values increased alongside the severity of fetal growth restriction and established a negative correlation between sFlt-1/PlGF ratio values and gestational age at the moment of delivery. Droge [43] showed, in a retrospective study, a similar pattern with a significant shorter time until delivery for high- and intermediate-risk patients with suspected pre-eclampsia based on sFlt-1/PlGF values (> 85 and 38–85, respectively) compared to low-risk women (cut-off < 38). Moreover, patients with adverse outcomes displayed significantly higher sFlt-1/PlGF values (177 vs. 14 for no adverse outcome). In a similar manner, Jeon [44] classified women with suspected pre-eclampsia into three groups based on sFlt-1/PlGF ratio: low-risk (< 38), intermediate-risk (38–85), and high-risk (> 85) women. The authors reported significantly lower gestational age at delivery among high-risk patients (32 weeks vs. 35.79 weeks for the low-risk group, *p* < 0.001), as well as a significantly higher prevalence of fetal growth restriction (< 10th centile: 75.6% vs. 10.5%, *p* = 0.023) and a significantly longer period of stay in the neonatal intensive care unit among the high-risk group (*p* = 0.003). Consequently, the authors regarded sFlt-1/PlGF as a useful indicator of pre-eclampsia severity and a reliable prognostic marker.

Ciobanu [45] found only a marginal and non-significant improvement in the prediction of delivery of a SGA neonate when adding the sFlt-1/PlGF ratio, mean uterine PI, and fetal MCA PI to fetal biometry and maternal factors at 35–37 weeks compared to the prediction performance only based on fetal biometry and maternal factors. 

An extensive study conducted by Gaccioli [46] reported a particularly strong association between the sFlt-1/PlGF ratio being > 5.78, as determined at 28 weeks, and pre-eclampsia with pre-term delivery in case of SGA infanta compared to pre-eclampsia with pre-term delivery in non-SGA infants (AUC 0.95 vs. AUC 0.66 at 95% CI). A strong association was obtained between sFlt-1/PlGF values > 38 measured at 36 weeks and term delivery of a SGA infant with maternal pre-eclampsia (AUC 0.95 vs. AUC 0.77 at 95% CI). Moreover, the combination of the sFlt-1/PlGF ratio > 5.78 and ultrasonic estimated fetal weight under the tenth centile doubled the positive predictive value of pre-term delivery of a SGA infant compared to screening only performed via ultrasound at 28 and 36 weeks.

In addition, regarding the sFlt-1/PlGF ratio and pre-term delivery, Heimberger [47] studied a population of patients diagnosed with chronic hypertension, reporting that women with elevated sFlt-1/PlGF ratio values (≥ 85) presented a significantly higher prevalence of pre-term delivery at < 34 weeks (40.5% vs. 7.7%, *p* < 0.001) and < 37 weeks (64.9% vs. 20.5%, *p* < 0.001), as well as significantly lower gestational age at delivery (34.7 vs. 38.2 weeks, *p* < 0.001), than women who displayed low sFlt-1/PlGF ratio values (<85). Moreover, women with sFlt-1/PlGF ≥ 85 were exposed to a significantly higher risk of developing superimposed or severe pre-eclampsia.

In a more recent paper, Gaccioli [48] found that the triggers responsible for increasing the sFlt-1/PlGF ratio are different in pre-eclampsia than fetal growth restriction: in pre-eclampsia, sFlt-1 displays increased concentrations in the placenta, leading to the elevated sFlt-1/PlGF ratio, whereas in fetal growth restriction, reduced placental expression of PlGF causes the elevated sFlt-1/PlGF ratio. Moreover, among patients already diagnosed with early fetal growth restriction (<32 weeks), Palma Dos Reis [49] demonstrated that sFlt-1/PLGF values > 85 measured when the diagnosis was established were associated with a shorter time until delivery (1.9 ± 1.52 weeks vs. 5.7 ± 3.2 weeks for sFlt-1/PlGF ≤ 85), as well as higher prevalence of fetal demise, albeit independently of pre-eclampsia.

Notably, sFlt-1/PlGF ratio < 38 demonstrated value in ruling out pre-term delivery within the next two weeks due to pre-eclampsia among twin pregnancies (median 98.9 for one week, 84.2 for two weeks vs. 23.5 for control, *p* < 0.001), having a negative predictive value of 98.8% for delivery within one week and a negative predictive value of 96.4% for delivery within two weeks from the assessment [50].

More insight into the ability of the sFlt-1/PlGF ratio to predict superimposed pre-eclampsia and adverse fetal outcomes was provided by Binder [51], who stated that women who developed superimposed pre-eclampsia within one week or later during pregnancy had significantly higher sFlt-1/PlGF values than women with chronic hypertension (114.5 for one week, 41.1 for later development of pre-eclampsia, and 4.7 for chronic hypertension, *p* < 0.001 for both). Moreover, women in whom sFlt-1/PlGF values were above the 97.5th centile were exposed to an increased risk of adverse perinatal outcomes, such as pre-term delivery at < 37 weeks or stillbirth (21.9% vs. 3.2% for control, *p* < 0.001).

The PEACOCK study [52], which was a recent major multicentric research including thirty six maternity units, proved that the angiogenic biomarkers did not help in the prediction of complications requiring delivery among patients diagnosed with late-onset pre-eclampsia (34–36 weeks and 6 days)—the reported negative predictive value was only 71.4%, while the specificity was 8.4%.

The main relevant studies that focus on the role of sFlt-1/PlGF ratio in the prediction and management of preeclampsia and other hypertensive disorders of pregnancy have been summarized in Table 1.

## 8. Relevant Data from the Recent Literature Regarding the Value of Doppler Velocimetry

In a cohort study of high-risk singleton pregnant women, Adekanmi [53] evaluated the role of uterine and umbilical artery PI, RI, PSV, end-diastolic velocity (EDV), and systolic/diastolic ratio (S/D) with the purpose of developing accurate prediction models for the identification of high-risk women who require appropriate interventions. The authors found significantly lower mean uterine EDV among women who developed pre-eclampsia than those who did not (25.97 cm/s vs. 34.96 cm/s, *p* = 0.003), significantly higher mean uterine RI in the pre-eclampsia group (0.59 vs. 0.5, *p* = 0.002) than in the group who did not, significantly higher mean uterine PI in those who developed pre-eclampsia (1.38 vs. 0.75, *p* < 0.001) than those who did not, and a significantly higher mean uterine S/D ratio in women who developed pre-eclampsia (2.79 vs. 1.92, *p* < 0.001) than those who did not. However, uterine PSV showed no significant differences. It was found that for a unit increase in uterine PI, the odds of pre-eclampsia increased 38.37 times. Moreover, uterine PI was proven to have a positive predictive value of 86% (AUC = 0.862, *p* < 0.001). As for predicting the severe form of pre-eclampsia, the combination of mean uterine PSV and mean umbilical PSV showed a positive predictive value of 80.3% (AUC = 0.8, *p* = 0.002). In addition, Trongpisutsak [54] found increased mean uterine PI at 16–24 weeks of gestation among patients who later developed pre-eclampsia (1.34 ± 0.52 vs. 0.98 ± 0.28 for control, *p* = 0.004)—the optimal cut-off value for mean uterine PI was 1.025—as well as higher prevalence of diastolic notching (45.5% vs. 11.2% for control, *p* < 0.001).

Soongsatitanon et al. [55] focused on the first-trimester prediction of pre-eclampsia via ultrasound examination of the uterine arteries at 11–13 weeks and 6 days. Authors reported significantly increased mean PI values among pregnancies later affected by pre-eclampsia (1.64 ± 0.59 vs. 1.46 ± 0.37 for the control group, *p* = 0.02). In the prediction of pre-eclampsia, uterine artery PI above the 95th centile showed 95.7% specificity and 92.3% negative predictive value. Similar results were presented by Oancea et al. [56], who reported a statistically significant difference of mean uterine PI at 11–14 weeks between patients who developed pre-eclampsia and the control (2.25 vs. 1.97, *p* = 0.012). The authors perceived screening for pre-eclampsia through uterine Doppler as particularly effective in healthcare systems with limited resources to perform the evaluation of other biomarkers. Additionally, in a multicentric study, Diguisto [25] found significantly higher mean PI (1.68 vs. 1.53, *p* = 0.05) and mean RI (0.74 vs. 0.7, *p* = 0.02), lower PI (1.54 vs. 1.21, *p* = 0.02), and higher prevalence of bilateral uterine notches (*p* = 0.01) at 11–13 weeks among women later affected by pre-eclampsia than the control group.

On the other hand, in a study conducted by Prakansamut [57], who also performed uterine artery Doppler screening at 11–13 weeks and 6 days with the purpose of predicting pre-eclampsia, no statistically significant differences in the mean uterine PI values (*p* = 0.66) or the presence of notching (*p* = 0.51) could be noticed between pregnant women who developed the condition and the control group.

Abdel Razik et al. [58] regarded an ultrasound Doppler scan at 20–24 weeks as one of the best predictors of pre-eclampsia. The authors reported significantly increased mean uterine artery PI and RI and higher prevalence of diastolic notch (*p* < 0.001) in pregnancies later complicated by pre-eclampsia and established cut-off values for PI (≥1.14), as well as for RI (>0.61).

Ratiu [59] focused on the role of uterine artery Doppler indices and notching measured at 19–22 weeks in the screening of adverse pregnancy outcomes, such as pre-eclampsia and intrauterine growth restriction in case of singleton pregnancy. Bilateral high RI and PI significantly increased both the prevalence of babies being small for their gestational age (neonates with birthweight <10th centile) and intrauterine growth restriction (birthweight < 3rd centile), whereas the presence of a notch significantly increased the prevalence of severe pre-eclampsia (4% vs. 0.6%, *p* = 0.008), HELLP Syndrome (2.4% vs. 0.2%, *p* = 0.05%), and oligohydramnios (5.7% vs. 0.7%, *p* = 0.004). Moreover, the presence of bilateral notch significantly increased the risk of SGA (64.4% vs. 26.6% with unilateral notch, *p* < 0.001), intrauterine growth restriction (49.2% vs. 6.3% with unilateral notch, *p* < 0.001), severe pre-eclampsia (5.1% vs. 3.1% with unilateral notch, *p* = 0.002), and HELLP syndrome (5.5% vs. 2.4% with unilateral notch, *p* < 0.001). Finally, bilateral uterine notching may represent a sign of babies being small for their gestational age and intrauterine growth restriction at the screening time.

Valuable insights into using Doppler evaluation of the uterine artery at 16–22 weeks of gestation for the prediction of adverse pregnancy outcomes was provided by Barati [60]. Abnormal uterine Doppler was defined as a mean PI > 1.45. The authors found reliable correlations between abnormal uterine Doppler at 16–22 weeks of gestation and the occurrence of early pre-eclampsia (<32 weeks), with a negative predictive value of 98.9%, positive predictive value of 88.2%, specificity of 95.5%, and sensitivity of 79% (*p* < 0.001). However, the ability of abnormal uterine Doppler to predict pre-eclampsia after 32 weeks was weaker (*p* = 0.025). Regarding other adverse obstetric outcomes, high prevalence of babies being small for their gestational age fetus (birthweight < 10th centile) in women with abnormal Doppler findings (23.5% vs. 0.82%), as well as pre-term delivery (11.8% vs. 1.4%), were recorded.

An extensive study including 6856 patients [61] who underwent uterine artery Doppler examination at 19–22 weeks highlighted higher median PI values in the case of women who developed pre-eclampsia or intrauterine growth restriction (1.36 vs. 0.93, *p* < 0.001), with the highest median PI recorded among patients who developed early-onset pre-eclampsia (1.68 vs. 1.31 compared to late-onset pre-eclampsia, *p* < 0.01). Consequently, Doppler screening of the uterine arteries at 20 weeks had the ability to identify 70.6% of cases with later development of early-onset pre-eclampsia and 73.3% of cases with early-onset intrauterine growth restriction. In addition, it was proved that early-onset pre-eclampsia had abnormal uterine Doppler PI (negative predictive value 99.9%), while in late-onset pre-eclampsia, only a fraction of patients displayed abnormal uterine Doppler results, thus suggesting that there are late-onset forms of pre-eclampsia with minimal placental involvement. Most notably, the authors of this study stated that uterine artery Doppler was an effective screening tool involved in the prediction of early-onset adverse outcomes that relate to impaired trophoblast invasion (pre-eclampsia or intrauterine growth restriction) in a population of unselected patients, without generally recognized risk factors for pre-eclampsia. This conclusion is in contrast to the recommendations of the other studies, which regarded uterine artery Doppler screening or sFlt-1/PlGF ratio determination as only being feasible for high-risk patients.

Moreover, Maged [62] showed that among uterine artery Doppler parameters screened at 18–22 weeks, mean uterine RI was significantly increased in patients who later developed pre-eclampsia (0.587 ± 0.072 vs. 0.524 ± 0.025 for control, *p* < 0.001) or intrauterine growth restriction (0.587 ± 0.053 vs. 0.524 ± 0.025 for control, *p* < 0.001). Pregnancies affected by pre-eclampsia and intrauterine growth restriction displayed significantly higher prevalence of unilateral or bilateral uterine artery notches. Medjedovic [63] also searched for ultrasound findings in the second trimester that could potentially increase the risk of pre-eclampsia. The unilateral notch of the uterine artery significantly correlated with later development of pre-eclampsia (47.62% specificity, 88.89% sensitivity, *p* = 0.023), while bilateral notch was an even stronger predictor of the condition.

In a recent retrospective study, Li et al. [64] assessed whether an abnormal uterine artery Doppler scan at 21–23 weeks of pregnancy increased the risk of pre-eclampsia. The authors reported that markers such as increased unilateral or bilateral PI (OR 2.36, 6.21, respectively) and increased bilateral RI (OR 2.83), unilateral, or bilateral notch (OR 3.66, 5.80, respectively) were effective tools for the prediction of pre-eclampsia. Moreover, for every 0.1 increase in the median multiple of mean PI, a 13% increase in the risk of pre-eclampsia could be noticed, and for every 0.1 increase in the median multiple of mean RI, the risk of pre-eclampsia was increased by 22%. 

In another recent study [65], Ekanem evaluated the role of Doppler examination of uterine arteries at 20–24 weeks of gestation in the screening of intrauterine growth restriction in a population with risk factors. Bilateral notching was found to significantly increase the risk of intrauterine growth restriction, low birth weight (*p* = 0.005), and low APGAR scores at 1 min (*p* = 0.007) and 5 min (*p* < 0.001). Babies born from pregnancies with bilateral uterine notching were also born at an earlier gestational age (*p* = 0.029). As a consequence, the authors pronounced themselves in favor of using uterine artery notching to appropriately manage pregnancies at high risk of intrauterine growth restriction.

Furthermore, Obican [66] focused on the performance of third-trimester uterine artery Doppler in predicting adverse pregnancy outcomes. Both left uterine artery notching and mean uterine PI > 95th centile were significantly correlated with babies being small for their gestational age (RR 1.76 and 1.83, respectively), pre-eclampsia (RR 2.53, 1.98, respectively), and early-onset pre-eclampsia (RR 2.88, RR 3.13). Chilumula [67] studied the correlation between uterine artery Doppler findings and adverse pregnancy outcomes in a population of women diagnosed with early- or late-onset severe preeclampsia. Early uterine diastolic notch was twice as likely to occur in the early-onset form than in late-onset preeclampsia(80% vs. 40%, *p* = 0.003). Abnormal uterine Doppler examination resulted in pregnancies with early-onset pre-eclampsia, increasing the risk of both maternal and neonatal complications, while among late-onset pre-eclampsia cases, abnormal Doppler predicted only perinatal complications. However, an abnormal Doppler scan failed to significantly correlate with caesarean delivery, gestational age at delivery, or other complications, such as eclampsia or HELLP syndrome.

Table 2 summarizes the main findings from above mentioned studies regarding the importance of uterine and umbilical Doppler parameters as predictors of preeclampsia and adverse perinatal outcomes.

## 9. Conclusions

sFlt-1/PlGF ratio ≤ 38 displays a high negative predictive value regarding ruling out PE within 4 weeks of assessment, i.e., between 24 and 37 weeks of gestation. In addition, the two angiogenic biomarkers can differentiate between mild and severe forms of pre-eclampsia, as well as gestational hypertension and pre-eclampsia. Mean uterine PI remains the first-choice Doppler parameter used to rule out PE, while the combination of uterine and umbilical PSV represents a useful tool that predicts the severity of PE. Uterine notching may be regarded as a predictor of severe PE, HELLP syndrome, intrauterine growth restriction, and oligohydramnios. Moreover, the sFtl-1/PlGF ratio is very effective at predicting early-onset PE with IUGR that requires delivery before 32 weeks and pre-term delivery in both early- and late-onset pre-eclampsia. Meanwhile, mean uterine RI and PI with notching are able to predict babies being small for their gestational age, intrauterine growth restriction, and pre-term delivery. sFlt-1/PlGF ratio and Doppler parameters prove superior performance compared to screening of PE based on clinical risk factors alone; therefore, the implementation of a standard screening based on angiogenic biomarkers and Doppler velocimetry may improve early detection of PE and other placenta-related disorders.

## Figures and Tables

**Figure 1 children-10-01430-f001:**
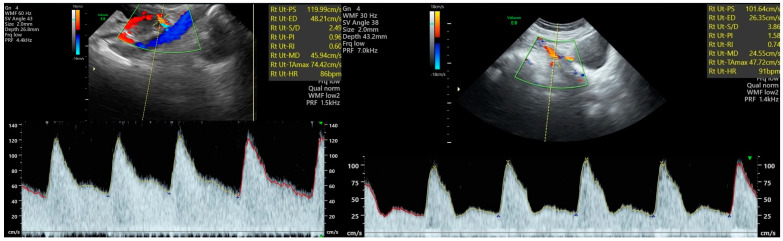
Transabdominal Doppler ultrasound examination of the uterine artery at 23 weeks of gestation. Left-normal pregnancy; PI value: 0.96 (within normal range). Right-pathological pregnancy in the case of a woman who later developed early-onset PE; severely increased uterine PI (1.58) and the presence of an early diastolic notch can be noticed. Legend for Figure 1: PE = pre-eclampsia, PI = pulsatility index.

**Table 1 children-10-01430-t001:** Data from the recent literature regarding sFlt-1/PlGF role in the prediction and follow-up of preeclampsia and adverse perinatal morbidity.

Nr.	Authors	Study Type	N	Objective	Conclusion
1	Verlohren, 2014 [23]	Prospective, multicentric	877	To establish gestational age-dependent cut-offs for the use of sFlt-1/PlGF ratio as a prognostic tool for PE	For early-onset PE, the sFlt-1/PlGF ratio ≤33 had the highest likelihood of a negative test, whereas values ≥ 85 had the highest likelihood of a positive test. For late-onset PE, the cut-offs were ≤ 33 and the rule-out and rule-in were ≥110 PE.
2	Zeisler, 2016—PROGNOSIS study [24]	Prospective, multicentric	1050	To assess if a low sFlt-1/PlGF ratio (at or below a cut-off) predicts the absence of PE within 1 week and whether a high ratio (above the cut-off) rules in PE within 4 weeks	sFlt-1/PlGF ≤ 38 ruled out PE within one week, regardless of gestational age, in women with clinically suspected PE (NPV 99.3%). sFlt-1/PlGF > 38 predicts PE occurrence within 4 weeks and a shorter time until delivery.
3	Bian, 2019—PROGNOSIS Asia study [25]	Prospective, multicentric	764	To assess the value of a sFlt-1/PlGF ratio for ruling out PE within 1 week To evaluate the predictive value of the ratio for fetal adverse outcomes	sFlt-1/PlGF ≤ 38 ruled out PE within one week with a similar performance as that of compared to PROGNOSIS study (NPV 98.6%). sFlt-1/PlGF was also a predicitve marker of PE at any time during pregnancy, as well as fetal adverse outcomes, shorter pregnancy duration, and pre-term delivery.
4	Zeisler, 2019 [26]	Prospective, multicentric	550	To assess the predictive value of the sFlt-1/PlGF ratio to rule out the onset of PE within 4 weeks in patients with suspected PE To assess the value of repeated measurements	In women with a sFlt-1/PlGF ratio of ≤38, PE can be ruled out for up to 4 weeks with a NPV of ≥94%. Retesting 2 or 3 weeks after the initial test improves risk stratification and decision-making.
5	Cerdeira, 2019—INSPIRE study [28]	Prospective, monocentric	370	To assess the sFlt-1/PlGF ratio’s ability to detect PE in daily clinical practice in cases of high-risk women	sFlt-1/PlGF ≤ 38 (low risk)/ > 38 (high risk) rules out/in PE within one week with better sensitivity and specificity than clinical practice alone, significantly improving clinical precision.
6	Caillon, 2018 [30]	Prospective, monocentric	67	To evaluate the routine use of the sFlt-1/PlGF ratio in a population of high-risk patients to predict PE	A cut-off < 38 for the sFlt-1/PlGF ratio is reliable for ruling out PE within 4 weeks (NPV 100%)
7	Perales, 2017—STEPS study [31]	Prospective, multicentric	819	To evaluate the sFlt-1/PlGF ratio as a predictive marker of early-onset PE in women at risk of PE	Early-onset PE: the sFlt-1/PlGF ratio measured between 20 and 28 weeks can improve prediction for women at risk. Late-onset PE: Only sFlt-1/PlGF measured at 28 weeks improves prediction.
8	Jeon, 2021 [44]	Prospective, monocentric	73	To assess sFlt-1/PlGF’s usefulness in terms of predicting adverse pregnancy outcomes in PE	sFlt-1/PlGF ≥ 85 (high-risk) predicted pre-term birth, reduced neonatal weight, and the need for neonatal intensive care.
9	Diguisto, 2017 [32]	Prospective, multicentric	226	To evaluate the accuracy of angiogenic biomarkers in the first trimester for PE screening in a high-risk population	PlGF and the sFlt-1/PlGF ratio are useful first-trimester markers involved in the prediction of PE.
10	Soundararajan, 2021—ROBUST study [33]	Prospective, monocentric	50	To evaluate if sFlt-1/PlGF may be used to predict the severity of PE among women at high risk	Patients with a sFlt-1/PlGF ratio > 85 carried a significant risk of developing severe PE associated with pre-term birth.
11	Tan, 2017 [34]	Prospective, monocentric	8063	To estimate at 31–34 weeks of gestation the patient-specific risk of PE via a combination of maternal characteristics and the sFlt-1/PlGF ratio To compare the performance of screening to that achieved only using the sFlt-1/PlGF ratio	Similar performance at 31–34 weeks in predicting delivery within the next four weeks due to PE between the combination and sFlt-1/PLGF alone. The combination displayed superior performance in the prediction of delivery due to PE four weeks after the assessment.
12	Ciciu, 2023 [36]	Prospective, monocentric	127	To assess the clinical utility of the sFtl-1/PIGF ratio in determining the diagnosis and severity of PE	sFlt-1/PlGF values may help differentiate between mild forms and severe forms of PE. sFlt-1/PlGF values can discriminate between PE and gestational hypertension.
13	Nikuei, 2020 [38]	Prospective, monocentric	58	To evaluate diagnostic accuracy of the sFlt-1/PlGF ratio for different forms of PE	The sFlt-1/PlGF ratio showed higher accuracy in terms of differentiating between PE and non-PE patients than for differentiating between severe or early-onset forms of PE.
14	Leanos-Miranda, 2020 [39]	Prospective, monocentric	810	To compare outcomes according to the degree of angiogenic imbalance, as assessed based on the levels of sFlt-1/PlGF in patients already diagnosed with PE (severe imbalance: ratio ≥ 85, mild imbalance: ratio 39–84, no imbalance: ratio ≤ 38)	Rates of pre-term delivery, delivery within 14 days, and delivery of a SGA infant were significantly higher among patients with severe angiogenic imbalance than patients with mild and no imbalance and among patients with mild imbalance vs. no imbalance. HELLP syndrome occurred only in the severe imbalance group.
15	Baltajian, 2016 [40]	Prospective, monocentric	100	To analyze sequential levels of plasma angiogenic factors among patients admitted due to PE	sFlt-1/PlGF ≥ 85 predicted a shorter time to delivery than sFlt-1/PlGF < 85 (6 days vs. 14 days). Significantly higher median sFlt-1/PlGF on admission for patients who presented adverse outcomes (205.9 vs. 47.5).
16	Peguero, 2021 [41]	Prospective, multicentric	63	To assess the potential influence of longitudinal changes in sFlt1/PlGF levels on the prediction of adverse outcomes among women with early-onset severe PE	Longitudinal daily changes in the sFlt-1/PlGF ratio correlated with a shorter time until delivery. Significantly increased daily changes in sFlt-1 values were recorded for those women who developed adverse outcomes: 1047 vs. 342 pg/mL/day.
17	Garcia-Manau, 2020 [42]	Prospective, monocentric	207	To compare sFlt-1/PlGF values and pregnancy outcomes among early-onset SGA/FGR stages	sFlt-1/PlGF values at diagnosis allowed the stratification of FGR severity.
18	Herraiz, 2018 [35]	Prospective, monocentric	5601	To analyze the usefulness of a clinical protocol for early detection of PE and FGR based on the measurement of the sFlt-1/PlGF ratio at 24–28 weeks of gestation	The sFlt-1/PlGF ratio and >95th centile, when measured at 24–28 weeks, were effective at predicting early-onset PE with FGR requiring delivery before 32 weeks.
19	Gaccioli, 2018 [46]	Prospective, monocentric	4098	To determine the effectiveness of sFlt-1/PlGF in predicting adverse pregnancy outcomes associated with the delivery of a SGA	A sFlt-1/PlGF ratio > 5.78 at 28 weeks was highly predictive of PE with pre-term delivery when the infant was SGA. sFlt-1/PlGF values > 38 at 36 weeks were predictive of the term delivery of a SGA infant with maternal PE.
20	Heimberger, 2020 [47]	Retrospective, monocentric	115	To compare characteristics and outcomes of women with chronic hypertension	sFlt-1/PlGF ≥ 85 increased the risk of pre-term delivery at <34 and <37 weeks, lower gestational age at delivery, superimposed PE, and severe PE.
21	Binder, 2020 [50]	Retrospective, monocentric	164	To evaluate the predictive value of the sFlt-1/PlGF ratio for delivery because of PE in twin pregnancies	sFlt-1/PlGF < 38 was useful in ruling out pre-term delivery due to PE.
22	Gaccioli, 2023 [48]	Prospective, monocentric	4212	To determine the relationship between maternal serum and placental levels of sFlt-1 and PlGF in women with a diagnosis of PE or IUGR	The contribution of sFlt-1 and PlGF to the increased sFlt-1/PlGF ratio are different in PE and IUGR: increased placental sFlt-1 imbalances the ratio in PE, whereas decreased PlGF imbalances the ratio in IUGR.
23	Palma Dos Reis, 2023 [49]	Prospective, monocentric	125	To evaluate if sFlt-1/PlGF ratio predicts faster fetal deterioration in early FGR	sFlt-1/PlGF > 85 predicted faster fetal deterioration, independently of PE, etc.
24	Binder, 2021 [51]	Retrospective, monocentric	142	To investigate the ability of sFlt-1/PlGF to predict superimposed PE or adverse pregnancy outcomes among patients with chronic hypertension	sFlt-1/PlGF significanlty improved the prediction of superimposed PE and adverse outcomes, such as stillbirth, pre-term delivery, etc.
25	Duhig, 2021—The PEACOCK study [52]	Prospective, multicentric	36	To assess the performance of sFlt-1/PlGF in predicting the need for delivery within seven days among women with late pre-term PE	In late pre-term PE, sFlt-1/PlGF did not add value to the clinical assessment.

Legend for Table 1: PE = preeclampsia, NPV = negative predictive value, SGA = small for gestational age, FGR = fetal growth restriction, IUGR = intrauterine growth restriction.

**Table 2 children-10-01430-t002:** Data from the recent literature regarding the role of Doppler velocimetry parameters in the prediction of pre-eclampsia and adverse perinatal outcomes.

Nr.	Authors	Study Type	N	Objective	Conclusion
1	Adekanmi, 2019 [46]	Prospective, monocentric	93	To develop accurate prediction models that identify women at high risk of PE and allow appropriate interventions	Cases that develop PE showed significantly lower uterine and umbilical PSV and EDV and higher uterine RI, PI, and S/D levels. Uterine PI is the best predictor for PE, while a combination of uterine and umbilical PSV predicted severity of PE.
2	Trongpisutsak, 2021 [47]	Prospective, monocentric	443	To assess if uterine artery Doppler at 16–24 weeks can predict PE	The optimal cut-off for PI was 1.025. Uterine diastolic notching also predicted PE.
3	Soongsatitanon, 2020 [48]	Prospective, monocentric	353	To determine the predictive value for PE using uterine PI in the first trimester	Uterine artery PI > 95th centile (2.17) in the first trimester was a marker of PE.
4	Oancea, 2020 [49]	Prospective, monocentric	120	To evaluate the potential of first-trimester uterine artery Doppler regarding early detection of PE in high-risk patients	Uterine PI in first trimester showed moderate predictive power.
5	Diguisto, 2017 [25]	Prospective, multicentric	226	To evaluate the accuracy of uterine artery Doppler in the first trimester for PE screening	Mean PI, lowest PI, mean RI, and bilateral notching were reliable first-trimester parameters in PE screening.
6	Prakansamut, 2019 [50]	Prospective, monocentric	405	To assess if uterine artery Doppler at 11–14 weeks can predict PE	No relevant PI values or presence of uterine notching in PE prediction.
7	Abdel Razik, 2018 [51]	Prospective, monocentric	270	To evaluate the role of uterine artery Doppler at 20–24 weeks in the prediction of PE	Cut-off values for mean PI (≥1.14), as well as mean RI (>0.61), were reported.
8	Ratiu, 2019 [52]	Prospective, monocentric	1472	To evaluate if uterine artery Doppler waveform analysis and the presence of a notch in the second trimester in unselected women with singleton pregnancies correlate with significant differences in common pregnancy outcomes	Bilateral high RI and PI and a notch significantly increased the development of SGA and IUGR. The presence of a notch significantly increases the development of severe PE, HELLP syndrome, and oligohydramnios. Bilateral notching was associated with IUGR or SGA at the screening time.
9	Barati, 2014 [53]	Prospective, monocentric	379	To investigate the predictive value of uterine artery Doppler in the identification of adverse pregnancy outcomes	Mean uterine artery PI > 1.45 measured at 16–22 weeks predicted an increased risk of PE, SGA, and pre-term delivery.
10	Llurba, 2009 [54]	Prospective, multicentric	6856	To examine the value of one-time uterine artery Doppler examination, performed at 20 weeks, in the prediction of PE and IUGR in a population of unselected patients	Doppler screening of the uterine arteries at 20 weeks was a feasible tool for the detection of pregnant women with a high risk for early-onset adverse outcomes, such as PE and IUGR.
11	Maged, 2017 [55]	Prospective, monocentric	453	To evaluate the role of uterine artery Doppler at 18–22 weeks as a predictor of PE and IUGR	Elevated uterine RI was a valuable marker that predicted PE (cut-off 0.55) and IUGR (cut-off 0.54).
12	Medjedovic, 2021 [56]	Prospective, multicentric	80	To investigate ultrasound risk factors for PE	Uterine artery notching (especially if present bilaterally) is a strong predictor.
13	Ekanem, 2023 [58]	Retrospective, multicentric		To evaluate the role of Doppler examination of uterine arteries at 20–24 weeks of gestation in the screening of IUGR in a population with risk factors	The implementation of uterine artery Doppler screening to predict high-risk IUGR pregnancies should include evaluation of notching besides mean PI at 20–24 weeks.
14	Obican, 2019 [59]	Prospective, monocentric	200	To assess the third-trimester uterine artery Doppler value in predicting adverse pregnancy outcomes in high-risk women	Left uterine artery notching and PI > 95th centile were associated with SGA, early-onset PE, and PE.
15	Chilumula, 2021 [60]	Prospective, monocentric	60	To correlate uterine artery Doppler results with maternal and neonatal outcomes in early- and late-onset severe PE	In early-onset PE, abnormal uterine Doppler results increased the risk of both maternal and neonatal complications. For late-onset PE, abnormal Doppler only predicted perinatal complications.

Legend for Table 2: PE = pre-eclampsia, PSV = peak systolic velocity, EDV = end-diastolic velocity, S/D = systolic/diastolic ratio, IUGR = intrauterine growth restriction.

## Data Availability

Not applicable.

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
