# Peer review of "The Current Role of the sFlt-1/PlGF Ratio and the Uterine–Umbilical–Cerebral Doppler Ultrasound in Predicting and Monitoring Hypertensive Disorders of Pregnancy: An Update with a Review of the Literature"

_children, 2023, doi:10.3390/children10091430_

Round 1

Reviewer 1 Report

The paper by Chirila et al. is a narrative review on the role of sFlt-1/PlGF and utero-umbilical doppler ultrasound in predicting hypertensive disorders of pregnancy. The paper is overall well written and organised, however I have few comments:

- The title is too generic and misleading. In fact Authors just focus on sFlt-1/PlGF and utero-umbilical arteries doppler ultrasound not on all biomarkers or ultrasound parameters. Title should be modified accordingly.

- Abstract is also too generic and should instead summarise the content and findings of the review. 

- In the Introduction section Authors report that: "Preeclampsia is the most frequent form of hypertensive disorder in pregnancy". However, Preeclampsia is the leading cause of maternal morbidity and mortality but not the most frequent form of HDP. Chronic hypertension and gestational hypertension are more frequent than PE, involving almost 15% of all pregnancies. 

- In the "Preeclampsia screening" section, instead of referring to "Clinical factors" only, I would refer to "Anamnestic factors" (e.g. previous history of HDP, first degree relative with HDP etc...)and "Clinical factors" (chronic hypertension, ART, BMI, etc...). Furthermore, among clinical factors I would mention serum uric acid (whose assessment during high-risk pregnancies has been recommended by international guidelines) and the novel serum uric acid to creatinine ratio, namely serum uric acid adjusted for kidney function, which has also recently been associated with an increased risk of PE. 

- In several sections throughout the text including conclusion, Authors stated that angiogenic markers and doppler ultrasound play a role in the diagnosis of HDP. I do not think this is appropriate since diagnosis of these disorders is made with specific clinical criteria. These markers help in predicting the incidence of these disorders and in defining the prognosis. They do not play any role in the diagnostic process.

- Authors stated "sFlt-1/PlGF ratio displays an almost perfect negative predictive value for ruling out preeclampsia within 4 weeks". I would temperate this sentence and overall all conclusions. First of all I would not use the word "perfect" since it has been shown that women can still develop PE without an increase in sFlt-1/PlGF ratio. Furthermore I would specify the timing of assessment. In fact the negative predictive value depends on the weeks of pregnancy in which the measurement is performed. The same comments are valid for the following paragraph on doppler ultrasound. 

- Tables should be reformatted and conclusions should be reduced to essential results only. Abbreviations should be used to facilitate the readability. 

Author Response

Point 1: The title is too generic and misleading. In fact Authors just focus on sFlt-1/PlGF and utero-umbilical arteries doppler ultrasound not on all biomarkers or ultrasound parameters. Title should be modified accordingly.

Response 1: TITLE: We changed biomarkers with sFlt-1/PlGF ratio as it is indeed more appropriate; We changed ultrasonography with uterine-umbilical-cerebral Doppler ultrasound as we also added a section regarding median cerebral artery Doppler in the pregnancy follow-up at the request of the second reviewer.

Point 2: Abstract is also too generic and should instead summarise the content and findings of the review. 

Response 2: Adjustements to the abstract have been made – values of sFlt-1/PlGF ratio and Doppler PI have been added.

Point 3: In the Introduction section Authors report that: "Preeclampsia is the most frequent form of hypertensive disorder in pregnancy". However, Preeclampsia is the leading cause of maternal morbidity and mortality but not the most frequent form of HDP. Chronic hypertension and gestational hypertension are more frequent than PE, involving almost 15% of all pregnancies. 

  • Response 3: We replaced Preeclampsia is the most frequent form of hypertensive disorder in pregnancy, with The incidence of preeclampsia is rising nowadays.

Point 4: In the "Preeclampsia screening" section, instead of referring to "Clinical factors" only, I would refer to "Anamnestic factors" (e.g. previous history of HDP, first degree relative with HDP etc...)and "Clinical factors" (chronic hypertension, ART, BMI, etc...). Furthermore, among clinical factors I would mention serum uric acid (whose assessment during high-risk pregnancies has been recommended by international guidelines) and the novel serum uric acid to creatinine ratio, namely serum uric acid adjusted for kidney function, which has also recently been associated with an increased risk of PE. 

  • Response 4: We introduced Anamnestic factors beside Clinical factors in the Preeclampsia screening section, as recommended; serum uric acid level and serum uric acid to creatinine ratio were also brought into discussion as clinical factors that increase the risk of preeclampsia.

Point 5: In several sections throughout the text including conclusion, Authors stated that angiogenic markers and doppler ultrasound play a role in the diagnosis of HDP. I do not think this is appropriate since diagnosis of these disorders is made with specific clinical criteria. These markers help in predicting the incidence of these disorders and in defining the prognosis. They do not play any role in the diagnostic process.

Response 5: Changes in the paper have been made to focus on the prognostic and screening value of the ratio and Doppler in predicting preeclampsia. However, the studies that discuss the diagnostic potential of sFlt-1/PlGF ratio (like the paper written by Ciciu and al) remained unchanged.  

Point 6: - Authors stated "sFlt-1/PlGF ratio displays an almost perfect negative predictive value for ruling out preeclampsia within 4 weeks". I would temperate this sentence and overall all conclusions. First of all I would not use the word "perfect" since it has been shown that women can still develop PE without an increase in sFlt-1/PlGF ratio. Furthermore I would specify the timing of assessment. In fact the negative predictive value depends on the weeks of pregnancy in which the measurement is performed. The same comments are valid for the following paragraph on doppler ultrasound. 

Response 6: We replaced almost perfect negative predictive value with high negative predictive value and we added between 24 and 37 weeks of gestation. Indeed almost perfect was too much. This conclusion is based on the research conducted by Zeisler et al.   

Point 7: Tables should be reformatted and conclusions should be reduced to essential results only. Abbreviations should be used to facilitate the readability. 

Response 7: Tables were slighlty reformatted to be more concise and conclusions have been reduced to essential content. Also, abbreviations have been put into place.

Reviewer 2 Report

Congratulations to the authors for an effective summary of the management of gestational hypertension.

Abstract: Although it is stated in the summary that information will be given about IUGR and SGA-like pathologies, the data on these pathologies are limited. In summary, it would be more accurate to concentrate on hypertension and even preeclampsia.

Figure: It is not clear whether the case sample of CDUS is pathological or normal, and what the pathological findings are. One normal CDUS image and a pathological case image with detailed description and marking of pathological findings should be placed next to it.

Radiologic evaluation: The CDUS examination method for the uterine artery and other vascular structures should be explained in a separate sub-title with a detailed description, even with the opinion of a radiologist, if possible.

Other vascular regions: A section should be created about the vascular structures evaluated in pregnancy follow-up, and normal parameters and abnormal values ​​should be summarized. For example fetal MCA etc.

Author Response

Point 1: Abstract: Although it is stated in the summary that information will be given about IUGR and SGA-like pathologies, the data on these pathologies are limited. In summary, it would be more accurate to concentrate on hypertension and even preeclampsia.

Response 1: We placed more focus on preeclampsia in the abstract. We cut such as gestational hypertension, intrauterine growth restriction (IUGR), small for gestational age (SGA) fetus, preterm birth was eliminated.

Point 2: Figure: It is not clear whether the case sample of CDUS is pathological or normal, and what the pathological findings are. One normal CDUS image and a pathological case image with detailed description and marking of pathological findings should be placed next to it.

Response 2: We added a normal uterine Doppler CDUS image alongside the pathological one, where we marked the pathological findings.

Point 3: Radiologic evaluation: The CDUS examination method for the uterine artery and other vascular structures should be explained in a separate sub-title with a detailed description, even with the opinion of a radiologist, if possible.

Response 3: We added a paragraph about the methodology of uterine Doppler, entitled Uterine artery Doppler – technique and reference parameters. We provided reference values.

Point 4: Other vascular regions: A section should be created about the vascular structures evaluated in pregnancy follow-up, and normal parameters and abnormal values ​​should be summarized. For example fetal MCA etc.

Response 4: Methodology and normal ranges of the umbilical artery and middle cerebral artery were explained in a newly created dedicated section - Other vascular structures evaluated in the pregnancy follow-up – Doppler technique and reference parameters.

Round 2

Reviewer 1 Report

Authors appropriately addressed my concerns and the paper is now suitable for publication